# Analysis of Gluten Content in Gluten-Free Pizza from Certified Take-Away Pizza Restaurants

**DOI:** 10.3390/foods7110180

**Published:** 2018-10-31

**Authors:** Daniela Manila Bianchi, Cristiana Maurella, Silvia Gallina, Ilaria Silvia Rossella Gorrasi, Maria Caramelli, Lucia Decastelli

**Affiliations:** 1Istituto Zooprofilattico Sperimentale del Piemonte, Liguria e Valle d’Aosta, Via Bologna 148, 10154 Torino, Italy; manila.bianchi@izsto.it (D.M.B.); cristiana.maurella@izsto.it (C.M.); silvia.gallina@izsto.it (S.G.); maria.caramelli@izsto.it (M.C.); lucia.decastelli@izsto.it (L.D.); 2Department of Public Health and Pediatrics, University of Turin, Piazza Polonia 94, 10126 Torino, Italy

**Keywords:** celiac, gluten-free pizza, gluten-free food, pizza restaurants

## Abstract

Currently, a strict gluten-free diet is the only treatment for celiac disease. In Italy, food service establishments and restaurants can be certified for providing gluten-free foods, including pizza restaurants that make both gluten-free pizza and traditional wheat-based pizza. With this study we analyzed the gluten content in samples of gluten-free pizza prepared and purchased at certified restaurants in the Turin metropolitan area. All samples, from 28 pizzas and 28 cooked dough bases, produced results below the test limit of detection, except for one sample of cooked dough, that tested positive for gluten but still below the warning level for celiac consumers (<20 ppm). Gluten-free pizza, as advertised in the restaurants surveyed, can be considered a safe option for gluten-free consumption. Attention to and compliance with good manufacturing practices, a requisite for obtaining gluten-free certification for restaurants, were noted to have a positive effect on the final product.

## 1. Introduction

Celiac disease is an immune-mediated disorder caused by the ingestion of gluten-containing foods in genetically susceptible persons [1]. Pathogenic mechanisms lead to inflammatory changes and altered architecture of the small intestinal mucosa, resulting in impaired nutrient absorption, diarrhea, and weight loss [2]. Now one of the most common lifelong disorders, celiac disease affects 1% of the population worldwide, with regional differences (e.g., 0.9–1.0% in Italy, 0.2% in Germany, 0.3–1.4% in Spain, 0.3–0.9% in the United States, 0.4% in Australia, and 0.5% in Brazil) [3], with a higher prevalence (10% to 15%) among persons with first-degree relatives with the condition [4]. A strict gluten-free diet is the only treatment for celiac disease: eliminating gluten-containing foods will usually resolve diarrhea and restore body weight and normal nutrient malabsorption. In addition, histopathological changes will also normalize [2].

To protect celiac and other food allergic consumers, European legislation requires the provision of allergen information on food labels so that sensitized consumers can avoid the foods that trigger their allergic reactions. Under European Regulation No 1169/2011 [5], the presence of allergens in a food product must be declared in the ingredients list on food labels. The gluten threshold is regulated by European Regulation 828/2014 [6] that went into effect in July 2016: the “gluten-free” claim can be made only if the food as sold to the final consumer contains no more than 20 mg/kg of gluten. Furthermore, the claim “very low gluten” can be made only if the food, consisting of or containing one or more ingredients made from wheat, rye, barley, oats, or their crossbred varieties specially processed to reduce the gluten content, contains no more than 100 mg/kg of gluten in the food, as sold to the final consumer.

Under European law, catering services, food retail businesses, and meal-serving systems must provide food allergen information. The Italian Ministry of Health has published a bulletin specifying how such allergen information should be made available to consumers [7]. The information can be displayed on menus, menu boards or signs, or any other equivalent system, including smart devices, so that it can be accessed easily and quickly. However, smartphone applications, bar codes, QR (Quick Response) codes, and similar means, cannot serve as the only information source, since they are not easily accessible to all consumers. The Health Ministry bulletin states further that a notice must be clearly displayed on the premises stating that “Information about the presence of substances or products that cause allergies or intolerances can be obtained from staff on duty” or “Information on substances and allergens will be provided verbally via waiting staff on request”.

The celiac population in Italy is estimated to be around 190,000 persons, according to the National Health Service registry [8]. A national association for celiac consumer protection is active in providing support, education, and advocacy for people with celiac disease, and assistance for food business operators in certifying gluten-free food products and food processes. The association has defined good manufacturing practices for pizza producers, conducts training courses, and has created a gluten-free certification for pizza restaurants that offer gluten-free pizza on the menu. The risk of gluten cross-contact is high in kitchens where both gluten-free pizza and traditional wheat-based pizza are produced, since aerosolized wheat flour can spread and contaminate gluten-free pizzas in any step of the production process [9].

The aim of this survey was to determine traces of gluten present in food products sold at certified gluten-free take-away pizza restaurants in the metropolitan Turin area.

## 2. Materials and Methods

### 2.1. Sampling Plan

The survey was conducted between June 2016 and June 2017. At the time of starting this survey, the public database of a national association for celiac consumer protection listed 32 take-away pizza restaurants as certified food service establishments: 12 located in the city of Turin and 20 in the metropolitan area. Staff members of CREALIA (Regional Center for Food Allergen and Intolerance), simulating celiac guests ordering one whole gluten-free take-away Margherita pizza collected samples from the pizza restaurants on a Saturday or a Sunday evening between 20:00 and 21:00, the busiest restaurant hours for pizzerias in Italy. The samples were kept closed in their delivery boxes and refrigerated (2–8 °C) until analysis, the following Monday morning, at the laboratory of the Food Safety Department, Istituto Zooprofilattico Sperimentale del Piemonte Liguria and Valle d’Aosta.

### 2.2. Gluten Analysis

Pizza Margherita is traditionally made with wheat flour, water, yeast, and salt, and then topped with tomato, mozzarella, olive oil, and other ingredients. Each sample was divided into two portions. One was analyzed as sold, and is referred to here as “complete Margherita Pizza”. The tomatoes and mozzarella topping from the other portion were removed, and only the cooked dough base was analyzed. This was done because the analytical result of the test is reported as a concentration: analyzing pizza dough as sample is useful to determine if the preparation of gluten-free dough is under control as a procedure; adding ingredients, such as tomatoes and mozzarella, that are prepacked and naturally gluten-free, could represent a dilution factor to the results of the complete pizza samples. Thus, two samples were obtained from each Margherita pizza: one was pizza as it is traditionally consumed, and the other was only the cooked dough base, this being the part that might most potentially be gluten contaminated. All samples were analyzed in two replicates for gluten detection and quantification using an ELISA Ridascreen^®^ Gliadin (R-Biopharm AG, Darmstadt, Germany) test, a sandwich enzyme immunoassay, using specific R5 antibodies against gliadins, for the quantitative analysis of prolamins from wheat (gliadin), rye (secalin), and barley (hordein) in gluten-free declared foods. According to the manufacturer’s instructions, the test’s limit of detection (LOD) is 0.5 (mg/kg) ppm of gliadin or 1 (mg/kg) ppm of gluten; the limit of quantification (LOQ) is 2.5 (mg/kg) ppm of gliadin or 5 (mg/kg) ppm of gluten. This test has been validated and accredited in the Food Safety Department, Istituto Zooprofilattico Sperimentale del Piemonte Liguria and Valle d’Aosta, and it is the one used for official food controls. Sample preparation and testing were performed by trained laboratory employees according to good laboratory practices using single use instruments and working under a chemical hood when required. The test manufacturer’s instructions were followed. In detail, 5 g of sample was homogenized, 0.25 g were transferred in a vial and mixed with 2.5 mL of Extraction Cocktail reagent. The vial was incubated for 40 min at 50 °C, and then mixed with 7.5 mL of 80% ethanol. After shaking for one hour with a rotator at room temperature, the vial was centrifuged 10 min at 2500× *g*, and the supernatant transferred in a screw top vial. The sample was diluted 1:12.5 with sample diluent and immediately used in the assay. Absorbance was read at 450 nm using a spectrophotometer iMark™ Microplate Reader BIO-RAD (Segrate, Italy), and the results were evaluated with onboard RIDA^®^SOFT Win software (R-Biopharm AG, Darmstadt, Germany). Samples with gluten level <20 ppm were considered gluten-free, as required by European legislation.

## 3. Results

Four of the 32 gluten-free certified pizza restaurants were not sampled: one reported that the gluten-free dough was finished on the evening of the sample collection, and the three others do not provide take-away service on the weekend. In all, 28 complete Margherita pizzas were collected from 28 pizza restaurants, yielding a total of 56 samples analyzed: 28 complete pizzas and 28 cooked pizza dough bases.

According to the result expression of the test method, samples can be negative (gluten <1 mg/kg, gliadin <0.5 mg/kg); positive with a low and not quantifiable concentration of the target protein (gluten between 1 and 2.5 mg/kg, gliadin between 0.5 and 5 mg/kg); positive with a concentration of target protein that can be quantified (gluten >5 mg/kg, gliadin >2.5 mg/kg).

All 28 complete Margherita pizza samples tested negative, and gave results below the test LOD of 1 mg/kg and 0.5 mg/kg for gluten and gliadin, respectively. Twenty-seven of the samples of cooked pizza dough bases resulted in being negative, and one sample tested positive, with a concentration of gluten and gliadin quantified as 10 and 5 mg/kg, respectively. As reported in Table 1, all samples were below the warning level of 20 mg/kg for celiac consumers.

## 4. Discussion

The risk of eating gluten-contaminated food when dining out is a genuine worry for persons with celiac disease [10]. In the United States, up to 25% of those having celiac disease for 2 to 5 years choose to not dine out at all [11]. Hidden gluten or cross-contamination in gluten-free products is considered as one of the possible causes of persistence of mucosal injury after 2 years of good adherence to a gluten-free diet in about 50% of a sample of celiac patients [12]. Cross-contamination can occur in the food production line, or at the time of cooking gluten-free foods at home, or when eating out, or when consuming ready-to-eat foods [13]. The risk of cross-contamination is higher in commercial kitchens where foods containing gluten-based ingredients and gluten-free foods are prepared.

Previous studies have chiefly investigated the gluten content in commercially available gluten-free foods (certified or naturally gluten-free products) [14,15,16,17,18,19] and, occasionally, in food declared being gluten-free or expected (e.g., beans), served in restaurants or sold through food outlets and shops producing and selling directly to the consumer [9,20,21,22].

A previous study investigating the gluten content in 84 gluten-free pizzas sampled from 5 pizza restaurants in Italy with co-manufacture of gluten-free and wheat-based pizzas [9] reported gluten content above the LOD in 9 pizza samples (10.7%) with levels always below 20 ppm; the results varied across the restaurants, however, probably because of differences in adherence to good manufacturing practices. The study also analyzed 70 gluten-free pizza samples prepared in the training kitchen of a traditional bakery school in Genoa, Italy, where a trained chef prepared wheat-based pizzas and gluten-free pizzas: the gluten content was above the LOD in 11 samples and above 20 ppm in one sample [9]. A very recent study of the risk for persons with celiac disease [20], reported that in 16 bakeries selling gluten-free products out of the 25 sampled in Brasilia, at least one sample tested positive for gluten contamination >20 ppm. A study conducted in self-service restaurants in Brazil showed that 16% of common beans collected and later analyzed via ELISA were contaminated by gluten [21]. In Australia, the city of Melbourne has conducted audits since 2014 on food businesses that advertise gluten-free options. The comparative data show an improvement in rates of non-compliance foods from 20% in 2014, to 15% and 9% in 2015 and 2016, respectively. The presence of detectable gluten in a gluten-free food implies a non-compliance for the Food Australia New Zealand (FSANZ) definition of gluten-free; the number of samples with gluten above 20 ppm presented for the 2016 audit was 6% [22].

Table 2 presents the results of our study and of others carried out in restaurants and other food businesses.

Educating chefs about gluten-free diet is one strategy to reduce gluten contamination in restaurant kitchens. In the United Kingdom in 2003, a study that evaluated chefs’ knowledge about celiac disease found that 322 of the chefs (161 restaurants, 161 take-away venues) interviewed were less knowledgeable about celiac disease than the public: less than one-fifth of the chefs surveyed in the study had reportedly ever heard of celiac disease [23]. A later follow-up study in 2013 found that awareness of celiac disease increased markedly, reaching 78% among the 265 chefs interviewed [10]. In the United States, 77% of 430 chefs interviewed in 2010 had heard of celiac disease [24] and, in New Zealand, all of the 90 chefs and cooks interviewed in 2013 were aware of the term “gluten-free diet”, but 13% were uninformed about celiac disease [25].

In our study, we found that the chefs in charge at the surveyed restaurants possessed good knowledge about celiac disease and gluten contamination. When asked to make a gluten-free pizza, they were willing and able to fill the order, assuming it came as a special request from a restaurant customer, rather than from a food safety survey requiring samples for laboratory analysis. It is important to note that all the pizza restaurants in the survey also make traditional pizza for non-celiac customers. Good manufacturing practices can be seen to have a positive effect on the final product. Indeed, the gluten-free certification program, conducted by the national association for celiac consumers’ protection, is designed to help food service establishments provide safe options for gluten-free consumers. To receive accreditation, food business operators must meet and adhere to best practice standards for all steps of food production and handling, from the choice and transportation of raw materials to the storage and production of ready-to-eat foods. The control measures regard work spaces, tools, surfaces, and the clothes worn by cooks and other staff while on duty. In general, the procedures apply to all food business operators, but some are specific for pizza restaurants. For example, to reduce the chance of cross-contamination, the ingredients for making gluten-free dough must be kept separate. The use of an oven only for baking gluten-free pizzas is recommended but, when absent, the same oven can be used for making both traditional and gluten-free pizza, as long as the two types are not baked at the same time and the gluten-free pizza is made in a high-sided baking pan. Furthermore, to mitigate the risk of cross-contamination, gluten-free flour should be spread over surfaces where all pizzas are prepared. Taken together, these strategies can help to raise consumer confidence that the pizzas purchased in certified restaurants is really gluten-free.

Among the limitations to this study is the fact that the authors did not sample four of the entire list of Pizza Restaurants certified by the Association for Celiac Consumers’ Protection in the Turin metropolitan area, the one-time sampling at each restaurant, and the fact that no formal statistical tests were applied.

## 5. Conclusions

Consumption of gluten-free pizza in the pizza restaurants surveyed in the present study can be considered safe for people with celiac disease. Fifty-five of the 56 samples analyzed (28 complete Margherita pizzas and 28 cooked pizza dough bases from 28 different pizza restaurants) tested negative for gluten, except one that tested positive with a concentration below the warning level of 20 ppm for celiac consumers. This concentration is the threshold limit the Codex Alimentarius states as being safe for consumption by persons with celiac disease. In addition, it is the legal limit for use of the claim “gluten-free food”. Inclusion in the public database of restaurants that can supply gluten-free food to consumers with celiac disease requires adherence to good manufacturing practices and training of operators. The results of this survey show that adherence to these procedures has a positive effect on the final product.

## Figures and Tables

**Table 1 foods-07-00180-t001:** Results of gliadin/gluten analysis of Margherita pizzas.

	Gluten <1 mg/kg;	Gluten >1 But <2.5 mg/kg;	Gluten >2.5 mg/kg;	Notes
	Gliadin <0.5 mg/kg	Gliadin >0.5 But <5 mg/kg	Gliadin >5 mg/kg	
Complete pizza samples (No.)	28	0	0	
Pizza dough base (No.)	27	0	1	Content:
				gliadin: 5 mg/kg;
				gluten: 10 mg/kg

**Table 2 foods-07-00180-t002:** Results of surveys on gluten content in declared or expected gluten-free foods from restaurants and other food businesses.

City	Survey Year	No. of Venues	Type of Venue	No. of Gluten-Free Products Sampled	Type of Food	No. of Samples with Gluten <LOD (%)	No. of Samples with Gluten >LOD and <20 ppm (%)	No. of Samples with Gluten >20 ppm (%)	Source
Turin (Italy)	2016–2017	28	Pizza restaurant	56	Gluten-free pizza	55 (98.2)	1 (1.8)	0	Present study
City not reported (Italy)	Not reported	5	Pizza restaurant	84	Gluten-free pizza	75 (89.3)	9 (10.7)	0	[9]
Brasilia (Brazil)	2014	25	Bakery	130	Gluten-free bread, biscuits and cakes; tapioca flour biscuits	Not reported	Not reported	28 (21.5)	[20]
Brasilia (Brazil)	Not reported	20	Restaurant	60	Beans	Not reported	Not reported	10 (16.6)	[21]
Melbourne (Australia)	2016	127	Food business	158	Mixed gluten-free foods (e.g., chicken burger, banana bread, calamari)	144 (91.1)	5 (3.2)	9 (5.7)	[22]

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
