# Peer review of "Analysis of Gluten Content in Gluten-Free Pizza from Certified Take-Away Pizza Restaurants"

_foods, 2018, doi:10.3390/foods7110180_

Round 1
Reviewer 1 Report
The improvement in the manuscript (foods-380857) entitled: “Pizza gluten-free: a survey of safety level for celiac consumers” is really evident. I congratulate the Authors for their determination and hard work to improve this article. In its present form, the manuscript is suitable and ready for publication as a brief report.
Author Response
Responses to Reviewer 1 Comments
The improvement in the manuscript (foods-380857) entitled: “Pizza gluten-free: a survey of safety level for celiac consumers” is really evident. I congratulate the Authors for their determination and hard work to improve this article. In its present form, the manuscript is suitable and ready for publication as a brief report.
Response: Thank you very much for your comments and suggestions which made possible the improvement of the manuscript.
Reviewer 2 Report
Opinion: I have gone through the revised version of the manuscript, that looks interesting that the previous version. My comments for the revised version are as follows:
Major Comments:
Comment #1: Line no 83-88; I am not fully convinced with the justification that toppings will disturb the ELISA dilution factor. The Ridascreen ELISA method recommends to homogenize the food sample before testing. If still, authors are confident about the dilution factor, they should write this point in the limitation of the study.
Comment #2: Line no. 93-99 (R5 ELISA process); I suggest to write in short the methodology of the R5 ELISA, a reader would like to know how the gluten was extracted from Pizza samples and what was the ELISA steps. If it is not feasible, I suggest to give reference of any published study, where the processing of gluten extraction and R5 ELISA was explained.
Comment #3: Line no 103-106; Authors have given a nice justification, I still have a small doubt. In the revised version, it is clear that study duration was one year. In the one year time, why the authors have not collected the dough samples from the restaurant where pizza dough was ended? If there is a valid justification it can also be mentioned as study limitation in the discussion part.
Comment #4: Line no. 101; please add the official cut off value of Ridascreen ELISA. (eg. Samples with < 20 ppm of gluten were considered as gluten free)
Minor comments:
Comment #1: The title, “Pizza gluten free is” somewhat not so self-explanatory word. This is not a comment but just a suggestion to make it “Gluten-Free Pizza” this is easily understandable.
Comment #2: There are some minor formatting errors, I suggest authors to check the formatting once. (eg. Line no. 18, 77)
Comment #3: Would it be better to start the line no. 128 “previous studies have…”?
Comment #4: Discussion: This section is nicely written, I suggest authors to write some of the limitations of the study as well.
Author Response
Response to Reviewer 2 Comments
Major Comments
Comment #1: Line no 83-88; I am not fully convinced with the justification that toppings will disturb the ELISA dilution factor. The Ridascreen ELISA method recommends to homogenize the food sample before testing. If still, authors are confident about the dilution factor, they should write this point in the limitation of the study.
Response #1: we did explain what is meant as dilution factor. Lines 88-91.
Comment #2: Line no. 93-99 (R5 ELISA process); I suggest to write in short the methodology of the R5 ELISA, a reader would like to know how the gluten was extracted from Pizza samples and what was the ELISA steps. If it is not feasible, I suggest to give reference of any published study, where the processing of gluten extraction and R5 ELISA was explained.
Response #2: we added the description of the extraction step of the sample. Lines 104-109.
Comment #3: Line no 103-106; Authors have given a nice justification, I still have a small doubt. In the revised version, it is clear that study duration was one year. In the one year time, why the authors have not collected the dough samples from the restaurant where pizza dough was ended? If there is a valid justification it can also be mentioned as study limitation in the discussion part.
Response #3: we added in the limitation of the study. Lines 214-217.
Comment #4: Line no. 101; please add the official cut off value of Ridascreen ELISA. (eg. Samples with < 20 ppm of gluten were considered as gluten free)
Response #4: we added, Lines 111-112.
Minor comments:
Comment #1: The title, “Pizza gluten free is” somewhat not so selfexplanatory word. This is not a comment but just a suggestion to make it “Gluten-Free Pizza” this is easily understandable.
Response #1: thank you, we used the word “gluten-free pizza”.
Comment #2: There are some minor formatting errors, I suggest authors to check the formatting once. (eg. Line no. 18, 77)
Response #2: done, thank you.
Comment #3: Would it be better to start the line no. 128 “previous studies have…”?
Response #3: thank you, we wrote “Previous studies have….”. Line 141.
Comment #4: Discussion: This section is nicely written, I suggest authors to write some of the limitations of the study as well.
Response #4: thank you, we added it, lines 214-21.
Reviewer 3 Report
Title: safety level is not surveyed. Instead gluten level in gluten-free pizza from certified take-away pizza restaurants is surveyed. Please change accordingly.
Line 90: Please indicate how many replicates were analyzed for each of the complete pizza and cooked dough base samples.
Lines 132-142 are not relevant to this study and could be removed since they apply to retail foods; and the pervious paragraph cites them anyway.
Line 162 is also not relevant since they prepackaged food is covered earlier and not focus of the study.
Line 205: ….for gluten except one tested….
Author Response
Responses to Reviewer 3 Comments
Title: safety level is not surveyed. Instead gluten level in gluten-free pizza from certified take-away pizza restaurants is surveyed. Please change accordingly.
Response: we propose this title: “Analysis of gluten content in gluten-free pizza from certified take-away pizza restaurants”.
Line 90: Please indicate how many replicates were analyzed for each of the complete pizza and cooked dough base samples.
Response: we added. Lines 93-94.
Lines 132-142 are not relevant to this study and could be removed since they apply to retail foods; and the pervious paragraph cites them anyway.
Response: we removed lines 132-144 (lines 143-155 of the revised manuscript version) with data referring to retail foods.
Line 162 is also not relevant since they prepackaged food is covered earlier and not focus of the study.
Response: we removed lines 162-163 (lines 145-157 of the revised manuscript version) and moved lines 164-165 (lines 176-178 of the revised manuscript version) at line 127 (line 138-140 of the revised manuscript version), where the subject is cross-contamination and the quote “The risk of cross-contamination is higher in commercial kitchens where foods containing gluten-based ingredients and gluten-free foods are prepared” results properly inserted. That sentence otherwise would have lose its meaning without the previous sentence (lines 162-163) that was removed.
Line 205: ….for gluten except one tested….
Response: modified, thank you, line 223.